# Local Lead–Lag Relationships and Nonlinear Granger Causality: An Empirical Analysis

**DOI:** 10.3390/e24030378

**Published:** 2022-03-08

**Authors:** Håkon Otneim, Geir Drage Berentsen, Dag Tjøstheim

**Affiliations:** 1Department of Business and Management Science, Norwegian School of Economics, 5045 Bergen, Norway; geir.berentsen@nhh.no; 2Department of Mathematics, University of Bergen, 7803 Bergen, Norway; dag.tjostheim@uib.no

**Keywords:** lead–lag relationships, local Gaussian approximation, local Gaussian autocorrelation, local Gaussian cross-correlation, local Gaussian partial correlation, test of conditional independence, nonlinear Granger causality test

## Abstract

The Granger causality test is essential for detecting lead–lag relationships between time series. Traditionally, one uses a linear version of the test, essentially based on a linear time series regression, itself being based on autocorrelations and cross-correlations of the series. In the present paper, we employ a local Gaussian approach in an empirical investigation of lead–lag and causality relations. The study is carried out for monthly recorded financial indices for ten countries in Europe, North America, Asia and Australia. The local Gaussian approach makes it possible to examine lead–lag relations locally and separately in the tails and in the center of the return distributions of the series. It is shown that this results in a new and much more detailed picture of these relationships. Typically, the dependence is much stronger in the tails than in the center of the return distributions. It is shown that the ensuing nonlinear Granger causality tests may detect causality where traditional linear tests fail.

## 1. Introduction

The autocorrelation, partial autocorrelation, and cross-correlation functions of a pair of time series play fundamental roles in classical Box–Jenkins analysis. In Chapters 6–9 of [1], it is discussed how these quantities can be used to identify the model within the class of ARIMA (Autoregressive Integrated Moving Average) models. In Chapters 10–11, on transfer function models, a (relatively simple) lead–lag modeling approach based on the cross-correlation for a pair of time series is introduced, with one series leading the other.

Ref. [2] used the same quantities in a time series regression in his derivation of a linear causality test, which subsequently was named after him as a “Granger causality” test. This concept has played an essential role in the causality analysis of time series, and we will return to a detailed description of it in Section 2. The subsequent body of literature has mainly been linear, again essentially based on the second-order concepts of autocorrelation and cross-correlation. An important and representative paper on predictability, covering monthly return data from several countries, is [3].

However, it is well known that the returns of financial time series possess properties that make them less suitable for linear correlation-based models; see, e.g., [4]. These properties are especially pronounced for daily data, but also present for monthly data, which is the type of data we will analyze in this paper. The returns are typically weakly autocorrelated, and their distributions possess heavy tails. In fact, as we will see later, autodependence is more prominent in the tails than in the center of the distribution. Such issues were partially addressed by introducing ARCH and GARCH models (see, e.g., [5] for a detailed description). In terms of correlation functions, the ARCH/GARCH series appear as white noise series, where there is no correlation from one time point to another. However, there can be strong statistical dependence, even long-range dependence, which is modeled in the volatility process rather than in the mean process.

It is obvious that the correlation concept is especially suited for Gaussian processes, where it gives a complete description of the dependence, from which linear Granger causality can also be tested using simple methods. Nevertheless, as indicated above, time series in finance, econometrics, and other fields are often non-Gaussian. To completely describe dependence in such time series, one needs to formulate dependence in terms of distributional properties. Considering lead–lag relationships and Granger causality in terms of such concepts is natural. Such an undertaking is the central theme of this paper. We will do this by using the recent formalism of local Gaussian approximation, which [6] statistical describe. This represents a way of retaining the Gaussian distribution as well as the autocorrelation and cross-correlation locally. It is then possible to preserve the elegance and simplicity of the traditional concepts, including a Box–Jenkins-type formalism, in a local distribution-like fashion. The local approach allows the extension of linear Granger causality to a nonlinear distributional framework, as is already indicated in Chapter 11 of [6].

This paper is organized as follows. We describe the methodology in some detail in Section 2, and illustrate some advantages over linear lead–lag modeling by a simple simulation example in Section 3. The main contribution of this paper is to use the local Gaussian approach in an empirical study of nonlinear lead–lag relationships and Granger causality for monthly stock return data from 10 countries and compare them to traditional linear analysis. We do this in Section 4, where we study the case of the US and UK in some detail and discuss a larger selection of countries as well. We refer to [6,7] for derivations of asymptotic properties of estimates of local quantities and theoretical properties of the nonlinear Granger causality test. We conclude in Section 5.

## 2. Methodology

### 2.1. The Local Gaussian
Correlation

The main idea is to approximate a general density *f* by a Gaussian density *locally*. In principle, one could approximate *f* by another family of distributions, as in [8], but the Gaussian has some truly amazing multivariate properties. Among them is the fact that the distributional properties, as far as dependence is concerned, are described completely by correlations between pairs of variables, so that for a *d*-dimensional Gaussian time series $\left\{X_{t}\right\}$, the dependence is described by the correlations of pairs $\left(X_{i,t},X_{j,t+h}\right)$, h≥0;i,j=1…,d.

More precisely, the idea of local Gaussian approximation can be outlined as follows: consider a general multivariate density *f*, and a point x in the domain of *f*. In a neighborhood of x, whose size is determined by a bandwidth b, *f* is approximated by a Gaussian distribution, which, in the bivariate case, can be denoted by $\psi (\mu_{1}\left(x\right),\mu_{2}\left(x\right),\sigma_{1}\left(x\right),\sigma_{2}\left(x\right),\rho \left(x\right))$, where μ,σ,ρ stand for mean, standard deviation, and correlation, respectively. Moving to another point y in the domain of *f*, one fits another Gaussian $\psi (\mu_{1}\left(y\right),\mu_{2}\left(y\right),\sigma_{1}\left(y\right),\sigma_{2}\left(y\right),\rho \left(y\right))$. If *f* itself is Gaussian with mean $\mu =(\mu_{1},\mu_{2})$, standard deviation $\sigma =(\sigma_{1},\sigma_{2})$, and correlation ρ, then the local approximations all coalesce with μ(x)≡μ, σ(x)≡σ and ρ(x)≡ρ.

One may ask how well our methodology works for global non-Gaussian distributions—for example, skewed distributions. It does work, but of course one may need more observations with use of a corresponding smaller bandwidth. For some practical experience with non-Gaussian data, we refer to [9,10], where, among several non-Gaussian distributions, there are a skewed *t*-distribution, a mixture distribution, and several copula-generated distributions.

In order to estimate the local parameters $\theta \left(x\right)=\left(\mu_{1}\left(x\right),\mu_{2}\left(x\right),\sigma_{1}\left(x\right),\sigma_{2}\left(x\right),\rho \left(x\right)\right)$ given *n* observations of $\left\{X_{t}\right\}$, we use the local-log likelihood of [8]. The aforementioned neighborhood is then determined by a kernel function *K* with a bandwidth b, such that
(1)$$L\left(\theta \left(x\right)\right)=n^{-1}\sum_{t=1}^{n}K_{b}\left(X_{t}-x\right)\log(\psi \left(X_{t},\theta \left(x\right)\right)-\int K_{b}\left(u-x\right)\psi \left(u,\theta \left(x\right)\right)\;du$$
where *K* is a nonnegative symmetric kernel function. We have used the standard normal density function as a kernel function throughout this paper. For a given point x of interest, maximizing (Equation 1) with respect to θ(x) results in a Gaussian approximation ψ of *f* in the neighborhood of x. The correlation ρ(x) in this approximation is termed the local correlation at the point x. In the case of a bivariate time series $\left\{X_{t}\right\}=\left(X_{1t},X_{2t}\right)$, the local autocorrelation at lag *h* is defined as $\rho_{ii,h}\left(x\right)$, i=1,2, which is the correlation of the Gaussian approximation of the density function $f_{X_{it},X_{i,t+h}}$ at the point x. Similarly, the local cross-correlation $\rho_{12,h}\left(x\right)$ is the correlation of the Gaussian approximation of $f_{X_{1t},X_{2,t+h}}$ at the point x. These local quantities, then, are the local autocorrelation and local cross-correlation of the time series $\left\{X_{t}\right\}$. They reduce to the ordinary autocorrelation and cross-correlation for a Gaussian time series. Some properties of these quantities are given in [11]; alternatively, see Chapters 6 and 8 of [6]. We typically analyze the local Gaussian correlation at diagonal points $x_{1}=x_{2}$, which gives us an instrument for looking at the local correlations at selected points of the range of $\left\{X_{t}\right\}$—for example, the 10th percentile, the 50th percentile, and the 90th percentile. These three points then represent the lower tail, the center, and the upper tail, respectively, of the bivariate distribution under consideration. This will be used in the examples in Section 4, and it will be seen that the dependence of the series is different at the extremes compared to the center values of the series, and we can even examine a concept such as Granger causality on a local scale: do we have Granger causality at the extremes, but not at the center, for instance? See discussion after Equation (Equation 4).

Partly for computational needs, it is often an advantage to change the marginal scale to a standard normal one; see, e.g., [11]. This can be achieved by the marginal transformations $Z_{it}=\Phi^{-1}\left[F_{i}\left(X_{it}\right)\right]$, i=1,2, where $F_{i}$ is the cumulative distribution function of $\left\{X_{i}\left(t\right)\right\}$, and Φ is the cumulative distribution of the standard normal. In practice, $F_{i}$ has to be replaced by the empirical distribution function ${\hat{F}}_{i}$. Subsequently, the local (and global) autocorrelation and cross-correlation can be studied in terms of $\left\{Z_{t}\right\}$, or more precisely in terms of $\left\{{\hat{Z}}_{t}\right\}$. This construct can be compared to the copula concept, where dependence is studied in terms of the uniform variables $U_{i}=F_{i}\left(X_{i}\right)$. We believe that it is advantageous to use a standard normal marginal scale to be better able to analyze dependence in the tails of the distribution; see, for instance, ref. [12] for a more thorough discussion about this. We will use the standard normal marginal scale in this paper, and the bandwidth b used below is determined on the *z*-scale. If desired, one can transform the location argument back to the marginal scale for the original observations.

### 2.2. Testing for Distributional Granger Causality

The technique of local Gaussian approximation can be used in a number of different applications, as witnessed by the various chapters in the [6], to which we refer for more precise and extended treatment. One such application is testing conditional independence and (distributional) Granger causality. Informally, the time series $X_{1,t}$ Granger causes $X_{2,t}$ if past values of $X_{1,t}$ are useful for predicting future values of $X_{2,t}$. More specifically, $X_{1,t}$ Granger causes $X_{2,t}$ if
(2)X2,t⊥I∗(t−1)∣IX1∗(t−1)
where ⊥ (⊥) denotes dependence (independence), and where $I^{*}\left(t-1\right)$ is all information available at time t−1 and $I_{X_{1}}^{*}\left(t-1\right)$ is the same information omitting the values of $X_{1,t}$ up to, but not including, time *t*. In practice, the hypothesis (Equation 2) cannot be tested in its full generality. However, by taking only effects up to the first lag into account, it is possible to formulate a sufficient (but not necessary) condition for (Equation 2), X2,t⊥X1,t−1∣X2,t−1, the converse of which constitutes a testable null hypothesis of first-order Granger noncausality:
(3)$$H_{0}:X_{2,t}⊥X_{1,t-1}|X_{2,t-1}.$$


There are several approaches for performing this test. The simplest is based on constraining the (possible) dependence to a first-order linear relationship such that
$$X_{2,t}=c+\phi X_{2,t-1}+\beta X_{1,t-1}+\epsilon_{t}$$
in which case (Equation 3) reduces to $H_{0}:\beta =0$, which is thus a test for conditional uncorrelatedness rather than conditional independence. In fact, if α denotes the partial correlation between two jointly Gaussian time series $X_{2,t}$ and $X_{1,t-1}$ given $X_{2,t-1}$, this is simply a test of $H_{0}:\alpha =0$.

In [7], a nonparametric test is proposed using a local version of the partial autocorrelation function α(x) defined locally at x. If one wants to test for Granger causality, one must test for distributional conditional independence in (Equation 3). There exist several conditional independence tests in the literature. Many of them are listed and explained by [7]. We have chosen to focus on a test involving the local partial correlation α(x), which can be defined analogously to the global partial correlation α, by means of local autocorrelations and cross-correlations. Instead of testing for α=0, one tests for α(x)≡0. Many more details are given by [7].

It may also be noted that the framework of mutual information and conditional mutual information can be used to test for, respectively, independence and conditional independence. See, for instance, the recent survey paper by [13].

The idea of using the analogy with the global Gaussian definition of partial correlation in conjunction with the already established local Gaussian correlation means that, in this way, α(x) describes the dependence between $X_{2,t}$ and $X_{1,t-1}$ given $X_{2,t-1}$ in a neigbourhood of $\left(X_{1,t-1},X_{2,t}\right)=\left(x_{1},x_{2}\right)=x$. Squared values of this quantity can be aggregated over a region *S* or over the entire domain to obtain a test statistic
(4)$$T=\int_{S}\alpha^{2}\left(u\right)d\hat{F}\left(u\right)$$
which can be used for testing region-wise or global conditional independence, the latter corresponding to (Equation 3). In this manner, one obtains a global nonlinear distributional test of Granger causality instead of a linear Granger causality test by choosing *S* large enough. By restricting *S*, one can test for Granger causality in a specific region—for instance, in a tail region. This test has power against several nonlinear types of conditional dependence and is indeed suitable for testing (Equation 3). Details of this are again given in [7], where this type of conditional independence test is also evaluated against other tests of (global) conditional independence, which in principle can be extended to nonlinear Granger causality tests. We will illustrate our test on the simulated example in Section 3, where the ordinary linear Granger causality test does not work, but where the test based on the partial local autocorrelation function does. In Section 4, which is the main part of our paper, we follow this up with a quite extensive analysis of empirical monthly stock return data from 10 countries, and where the nonlinear as well as the linear Granger causality test, are applied to each pair of countries. It will be seen that much more information can be gained from the local autocorrelations and cross-correlations, and that this has consequences for the causality tests as well.

## 3. A Simulation Example

We motivate the empirical analysis that follows in Section 4 with a simulation experiment. Consider the following bivariate ARCH model in terms of conditional variances and conditional covariances $\sigma_{ij,t|t-1},i,j=1,2$,
(5)$$X_{t}=\Sigma_{t|t-1}^{1/2}\epsilon_{t}={\left(\begin{array}{cc} \sigma_{11,t|t-1} & \sigma_{12,t|t-1}\\ \sigma_{21,t|t-1} & \sigma_{22,t|t-1} \end{array}\right)}^{1/2}\epsilon_{t},$$
where $\epsilon_{t}∼N\left(0,I\right)$, and
$$\mathrm{vech}\left(\begin{array}{cc} \sigma_{11,t|t-1} & \sigma_{12,t|t-1}\\ \sigma_{21,t|t-1} & \sigma_{22,t|t-1} \end{array}\right)=\left(\begin{array}{c} \sigma_{11,t|t-1}\\ \sigma_{12,t|t-1}\\ \sigma_{22,t|t-1} \end{array}\right)=\left(\begin{array}{c} \gamma_{10}\\ \gamma_{20}\\ \gamma_{30} \end{array}\right)+\left(\begin{array}{ccc} \gamma_{11} & \gamma_{12} & \gamma_{13}\\ \gamma_{21} & \gamma_{22} & \gamma_{23}\\ \gamma_{31} & \gamma_{32} & \gamma_{33} \end{array}\right)\left(\begin{array}{c} X_{1,t-1}^{2}\\ X_{1,t-1}X_{2,t-1}\\ X_{2,t-1}^{2} \end{array}\right).$$
See, for instance, [14], Chapter 8, for more details about this model, including stationarity conditions. If we set
$$\left(\begin{array}{c} \gamma_{10}\\ \gamma_{20}\\ \gamma_{30} \end{array}\right)=\left(\begin{array}{c} 0.1\\ 0\\ 0.1 \end{array}\right)\;\mathrm{and}\;\left(\begin{array}{ccc} \gamma_{11} & \gamma_{12} & \gamma_{13}\\ \gamma_{21} & \gamma_{22} & \gamma_{23}\\ \gamma_{31} & \gamma_{32} & \gamma_{33} \end{array}\right)=\left(\begin{array}{ccc} 0 & 0 & 0\\ 0 & 0 & 0\\ 0.9 & 0 & 0.9 \end{array}\right),$$
we get a bivariate ARCH process, where the first component $X_{1}$ clearly Granger causes the second component $X_{2}$, but where there is no Granger causality in the other direction. The serial and cross-dependence in this case are nonlinear and not visible in the usual autocorrelation and cross-correlation plots; see the top row of Figure 1 for an example with n=10,000 observations. The corresponding plots for the squared process are presented in the second row of this figure, where we see that there is strong autocorrelation in the second component (which is typical for ARCH series), but there is still no significant cross-correlation from the first component, from which we can infer the influence from the first to the second component.

One way to detect such relationships is to consider the properties of specific parts of the joint distribution of the two components—for instance, using the local Gaussian approach that we sketched in Section 2. Specifically, in order to calculate the local cross-correlation between $X_{1,t}$ and $X_{2,t-k}$ at some quantile, we estimate the joint density of $\left(X_{1,t},X_{2,t-k}\right)$ using a local Gaussian approximation at that point and extract the local correlation from this fit. Doing this for various values of the lag index *k* and at three different percentiles (the 10th, 50th, and 90th quantiles of the two marginal components) results in the estimates seen in Figure 2. In all three plots, we have used the bandwidth b=1 in order to obtain the estimates. We see that there is no particular evidence for lagged cross-dependence between the two components of the bivariate process when looking at the center of this distribution. However, at the low and high quantiles, there are clear indications of component 1 leading component 2.

The local Gaussian correlation enables us to perform a much more detailed analysis of the joint distribution of the two time series components at various time lags. By fitting the normal distribution locally at the three different points along the diagonal $x_{1}=x_{2}$ of the sample space—the 10th, 50th, and 90th percentiles—we see that the lead–lag relationship (which clearly exists between the two components) is only visible at the two extremes, and not detectable in the center of the distribution. Confidence intervals for the local quantities in this section and in Section 4 can be obtained using the block bootstrap, as demonstrated in [11]. Since they are not used explicitly in the testing, they are omitted here; in Section 4, we refer instead to the formal tests themselves, where we obtain *p*-values using the bootstrapping scheme by [7].

We have also carried out this experiment with $\left\{\epsilon_{t}\right\}$ consisting of independent t(5)-distributed variables. The *t*-distribution is often used in practical ARCH/GARCH-type modeling. The results are practically identical to the case with Gaussian innovations, but not shown here.

In the following section, we will apply this technique to observed series of stock return data, as well as a more formal test for nonlinear Granger causality.

## 4. Lead–Lag Relations for Global and Local Correlations

In this section, we investigate lead–lag relationships among monthly country stock returns. Let $X_{t}$ be the value of the stock at time *t*. The return at time *t* is defined by $r_{t}=\log X_{t}/\log X_{t-1}$. The return series $\left\{r_{t}\right\}$ is assumed to be stationary, which is a standard assumption in the analysis of stocks. All of the local analysis, the result of which is displayed for instance in Figure 3, is based on this stationarity assumption since the concept of the local Gaussian correlation itself is based on this assumption. We have obtained all available data for each country from [15], up to and including the observation made on 1 November 2021; see Table 1 for details. A comprehensive study of lead–lag relationships among monthly country stock returns using traditional Granger causality can be found in [3]. This is partly the reason that we have chosen to analyze monthly data. In fact, the nonlinearity effects are even more pronounced for daily data, but in the present investigation, we wanted to avoid the issue of time difference in opening hours for the various stock exchanges, which, if left unadjusted for, may lead to problems of interpretation for lead–lag relations. It should be noted, however, that the study by [3] is not directly comparable to ours. Firstly, the time interval is different. For instance, there are twice as many observations in our investigation for the US–UK pair. Secondly, they consider returns in excess of the risk-free monthly rate for each country, while we only consider the raw return series. Finally, they control for dividend yield and a nominal interest rate in their model.

### 4.1. The Case of the US and the United Kingdom

Let us first consider one pair of countries in some detail. We have found the lead–lag relationships between the main stock indices of the US (S&P500) and the UK (FTSE 100) to be particularly rich in detail. We have n=196 monthly total market returns on these time series, starting in August 2005 (see Table 1 for details on the data set that we analyze in this paper).

We start by considering ordinary autocorrelation and cross-correlation plots, which we display in the top row of Figure 3. Following the discussion in Section 1 and Section 3, we are not surprised to see that there is very little serial and cross-dependence to be seen in these data. The UK series appears to have a first-order linear autoregressive correlation structure. However, there seems to be no autocorrelation of any statistically significant magnitude in the American series (95% confidence intervals are displayed). Moving on to the cross-correlations between the two series, a few lags have statistically significant linear correlations in either direction, but the signal as a whole is very weak in this figure.

Next, we consider the bivariate distributions in question in more detail using the local Gaussian correlation. In rows 2–4 in Figure 3, we see the corresponding *local* auto- and cross-correlations for these two time series calculated at the 10%, 50%, and 90% percentiles using the bandwidth b=1. We see a much stronger pattern in the local plots compared to the global plots. At the 10% percentile, there is fairly strong autodependence in the UK time series. This picture is consistent with the stylized facts of financial time series: even though there is little correlation in the global linear sense, there are typically still strong and important dependence structures to be found. See also [11]. We see a similar tendency in the US series, albeit perhaps not as strong as in the UK series. For the local cross-correlations on the 10th percentile, we first see a strong co-temporal local dependence between these two series (lag k=0) and indications of lead–lag patterns in both directions. The first three lags of correlation in the direction where the UK leads the US (lags k=−1,−2 and −3) are fairly strong, as well as the two first cross-correlations in the other direction (k=1,2). It should be carefully noted that the lagged cross-correlations are also influenced by the autodependence of the two series. The Granger causality test does not suffer from this since it is, in essence, a conditional independence test. Nevertheless, it is satisfying, as will be seen later in this section, that the informal indications of lead–lag relations that can be gleaned from Figure 3 are consistent with the more formal results of the Granger causality test of Table 2.

At the 90th percentile, we see that mainly the American series shows a marked dependence structure different from the corresponding global series. Moreover, the one-lag cross-correlation from the US to the UK is much stronger than vice versa.

At the 50th percentile, however, there is a lesser dependence structure, which we see by comparing the plots to the corresponding global plots.

We can investigate the local correlation structure for this country pair in more detail by plotting the local Gaussian correlation as a function of empirical percentiles for a few selected lags; see Figure 4. In the UK series, we see again that the most substantial autodependence can be found in the left tail, as measured along the diagonal. The interpretation of this finding is that there is a clear tendency that a series of significant negative returns depend on each other to a much higher degree than a series of observations closer to zero. For higher percentiles, we see that the autodependence at lags 1 and 3 increases somewhat, but also that the autodependence at lag 2 continues to decrease and becomes negative. This is a somewhat atypical behavior, as we will see in the following sub-section.

The US series display a more typical pattern in the local autocorrelations. At the three first lags, we see a “bathtub” shape, with higher local dependence at the extremes (where the lower extremes contain stronger dependence than the upper extremes) and less local dependence in the center of the distribution.

The cross-correlations show similar patterns. There is a strong co-temporal dependence between the two series, as seen in the red line. Moreover, the local cross-correlations are generally stronger at the extremes of the respective distributions than in the center. These observations are consistent with the stylized fact about financial time series wherein the dependence increases as the market goes up and, particularly, down.

However, it is important to reiterate and keep in mind that although we consider local correlations that enable us to detect nonlinear patterns in the dependence structure of the return distributions, the analysis above is still mainly descriptive. Conditioning as in the Granger causality test is needed to obtain more reliable and formal notions of lead–lag relations and causality. In order to proceed in this direction, we now consider distributional Granger causality, as described in Section 2.2.

The results of both the linear and distributional Granger causality test are displayed as part of Table 2. We refer to Section 4.2 for a more detailed description of this table. For the US and the UK, we see that the first-order *linear* test for Granger causality does not reject the null hypothesis of Granger noncausality from the UK to the US, as well as from the US to the UK. The corresponding test for distributional Granger noncausality rejects both hypotheses at the 5% significance level, indicating that the causality structure between these two time series manifests itself differently than through the global correlation structure. As mentioned in Section 1, a well-known phenomenon of financial time series is that global correlations, in general, fail in describing the dependence. See [16] for a different approach to this problem. Note that Granger causality could be present at higher lags; see [7].

### 4.2. A Wider Selection of Countries

We can now move on to examine the lead–lag relationships between a larger selection of monthly stock return time series using the methods from earlier sections. We consider the total market return on the primary stock exchange in 10 countries of various sizes and importance; see Table 1 for details on the origin and length of the individual series. All the data have been obtained from [15].

There are many plots to consider in this section, and they have been placed in a separate figure section at the end of the paper. The global autocorrelations for all ten series can be found on the diagonal of the matrix of plots in Figure 5. The global cross-correlations can be found in the plots in the off-diagonal in the same figure. We see the same general pattern that we recognize from the top row of Figure 3: the global autocorrelation and lagged cross-correlation signals in all of these series are very weak and hardly detectable. We note (see also Figure 3) that the time series of returns of the UK FTSE 100 Index differs from most of the other series in our data set, in that there is a clear first-order autoregressive effect that is significantly different from zero. We keep this in mind because we will see later in this section that this is the only country in our sample for which its leading effects on other countries are primarily detectable using the classical linear test for Granger causality. We see from the cross-correlation plots that the UK series is the only one that systematically indicates statistically significant linear lead and/or lag effects with several other countries.

In Figure 6, we see the corresponding plots of local auto- and cross-correlations for all (pairwise combinations of) countries, with the local versions calculated on the 10th percentile along the diagonal $x_{1}=x_{2}$. These illustrations correspond to the second row of plots in Figure 3. The general picture is that the local correlations in this part of the distribution are stronger than the corresponding global correlations (we will return to a formal statistical inference using the Granger causality test at the end of this section). The cross-correlations, in particular, are prominent in this set of plots, and we see the strongest set of lead–lag relationships between the continental European countries (Switzerland, Germany, France, and The Netherlands), and a somewhat weaker relationship between these countries and the UK.

In the center of the distributions, represented in Figure 7, where we plot the local auto- and cross-correlations at the 50th percentile along the diagonal, we return to a picture that is more similar to the global counterpart in Figure 5, although the signal is stronger, which may indicate that the local approach to a larger degree is capable of detecting (G)ARCH-type dependence structures in the same manner as we demonstrated in Section 3. Another interesting case in this figure is the UK, whose local correlations appear weaker than its global correlations, and we will return to this at the end of this section.

We can also inspect the local correlations at the 90th percentile as seen in Figure 8, which are stronger than at the 50th percentile but generally weaker than the 10th percentile.

In the same way as in the previous sub-section, we can analyze the local correlations by plotting some of them as a function of the percentile on the diagonal $x_{1}=x_{2}$. We see the result of this exercise for all countries and all country pairs in Figure 9, and this set of plots gives a more complete picture of the local dependence structure between the countries. The local autocorrelations tend to have similar patterns for the three first lags. The general trend is that the local correlation is lowest around the 50th percentile, and that it tends to increase more as the percentile decreases towards zero than when it increases towards 100. There is a clear tendency for the local correlation curves to have a somewhat asymmetrical U shape. We see strong correlations again within the continental European block. The US and Japan have a strong U shape in their cross-correlations, indicating that they follow each other when the market is going up and when it is going down. The cross-correlation curves for most country pairs are similar for positive and negative lags, indicating that the lead–lag relationship between these countries is symmetrical.

In Table 2, we see the results for the testing of the hypothesis of first-order Granger noncausality. The significance codes of this table were obtained from *p*-values using the simulation procedure of [7]. Each cell in the table reports the result from the causality tests *from* the country indicated by the row, *to* the country indicated by the column. The top symbol represents the result from the linear test, and the bottom symbol represents the result from the nonlinear test based on the local Gaussian correlation. We recognize, for instance, the case of the US and the UK from the previous sub-section, where the nonlinear test rejects the null hypothesis of Granger noncausality in both directions, while the linear test does not.

There are at least two interesting insights to derive from this table. We note first that the causality signals between these time series are generally weak. There are many more non-rejections than rejections of the causality tests, and this is true for both testing methods. Indeed, the linear test rejects 11.1% of the hypotheses, while the corresponding number for the nonlinear test is 15.6%. In Figure 10, we visualize the test results by plotting the *p*-values of the two different methods in a scatter diagram along with their marginal histograms. The *p*-values for the linear test display an almost uniform behavior, which we expect if the true value of the regression coefficient in question is equal to zero in all cases. The nonlinear test does produce more rejections of Granger noncausality than the linear test, and this trend is easy to see when considering the marginal histogram of *p*-values in Figure 10. There is a clear tendency in the data set that the causality signal in many cases is stronger from the nonlinear perspective, albeit not always individually statistically significant at the typical 5% level (indicated as vertical and dashed lines in the figure). Indeed, we cannot reject the null hypothesis that the *p*-values resulting from the linear test are drawn from the uniform distribution (p = 0.23 using a standard $\chi^{2}$ test). From a multiple comparison point of view, this may indicate that the linear rejections of noncausality are spurious. The same $\chi^{2}$ test is rejected for the *p*-values resulting from the nonlinear test based on the local Gaussian correlation (p = 2.5e-10).

Secondly, focusing on individual pairs, we see that the UK time series to a large degree Granger causes time series from other countries in a *linear* fashion. We have indicated these cases with a red circle in Figure 10. We also see that these relationships are generally not detected by the nonlinear test, which is naturally less powerful than the linear test in cases where the linear specification is good. We see a similar behavior in the tests for Granger noncausality from other countries *to* Norway, which is the smallest economy in the data set. These results are indicated in Figure 10 with blue circles.

## 5. Conclusions

In this paper, we have carried out an empirical investigation of monthly financial indices in 10 countries from Europe, North America, Asia, and Australia. This has been done using the approach of a local Gaussian approximation that has been explored in the book by [6] and in several recent papers. Such a point of view makes it possible to obtain an indication of lead–lag relations locally using separate analyses in the tails and in the center of the financial return distributions. Moreover, it makes it possible to employ corresponding nonlinear Granger causality tests. Ours is fundamentally a distribution-based analysis, which is entirely different from the traditional linear second-order one. The local investigation reveals that the autocorrelation for a return series and the cross-correlation between two return series are much more pronounced in the tails of the distributions, i.e., when the market is going up or down, than it is in the center, where the market is stable. Moreover, we are able to determine these differences quantitatively. The results are consistent with a nonlinear Granger causality test based on the local partial correlation. We have demonstrated that this test can pick up causality relations not detected by the linear test. The investigation indicates that causality may advantageously be studied locally using distributional concepts rather than globally, as traditionally done.

## Figures and Tables

**Figure 1 entropy-24-00378-f001:**
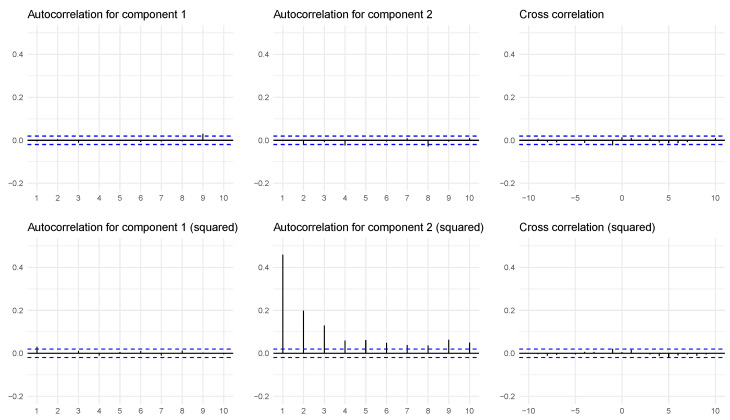
(**Top row**): Empirical autocorrelations for the two components in the simulated bivariate ARCH example, as well as the cross-correlation between the two series. (**Bottom row**): Corresponding plots for the squared series. Error limits are 95% confidence intervals. The *x*-axis indicate lags, and the auto- and cross-correlations are mapped to the *y*-axis.

**Figure 2 entropy-24-00378-f002:**
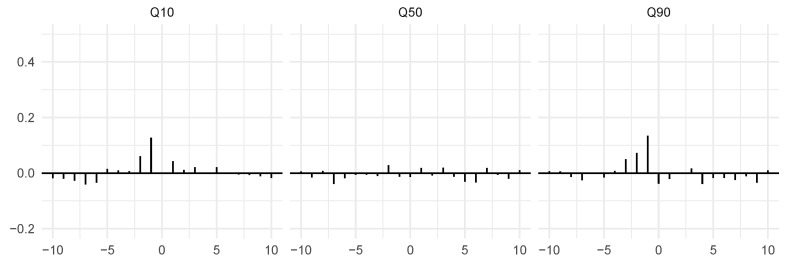
Local cross-correlations for the pair $(X_{1,t},X_{2,t-k}$ with *k* ranging from −10 to 10 for the simulated example at the 10th, 50th, and the 90th percentiles, calculated using bandwidth b=1. There is clear evidence that the first component leads the second, but this is only visible at the lower (Q10) and upper (Q90) ends of the diagonal $x_{1}=x_{2}$ of the sample space. The *x*-axis indicates lags, and the cross-correlations are mapped to the *y*-axis.

**Figure 3 entropy-24-00378-f003:**
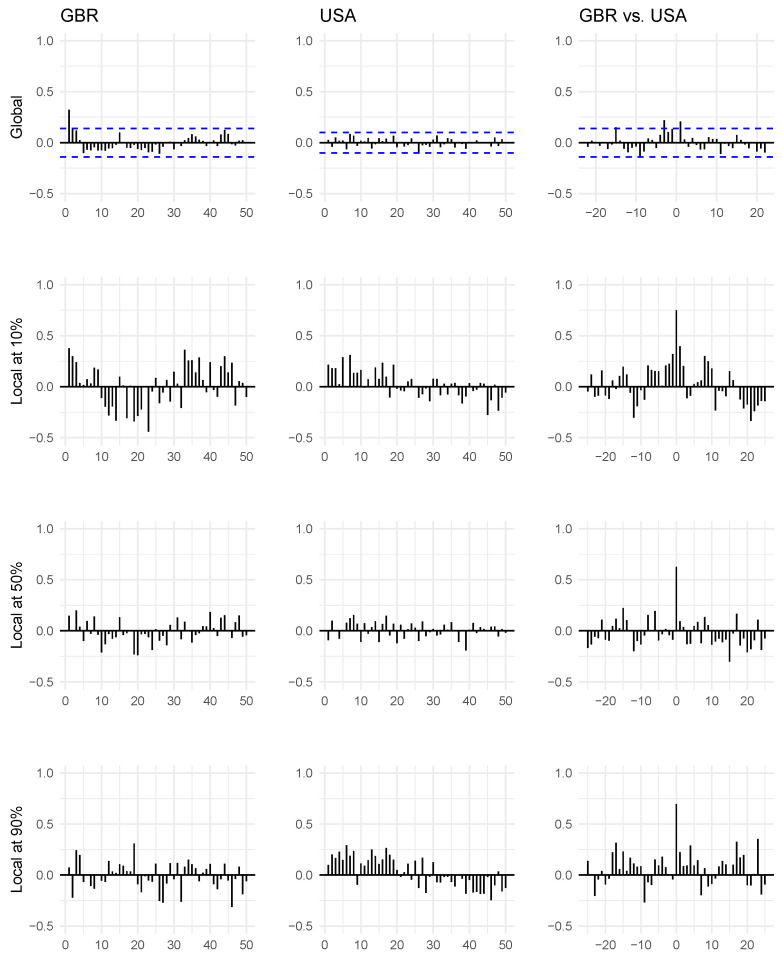
Autocorrelation and cross-correlation plot for the US and the UK. The top row displays ordinary correlations, while the last three rows display the local Gaussian correlation for the 10%, 50%, and 90% percentiles using the bandwidth b=1. Negative lags correspond to the UK leading the US, positive to the US leading the UK. The *x*-axis indicates lags, and the auto- and cross-correlations are mapped to the *y*-axis.

**Figure 4 entropy-24-00378-f004:**
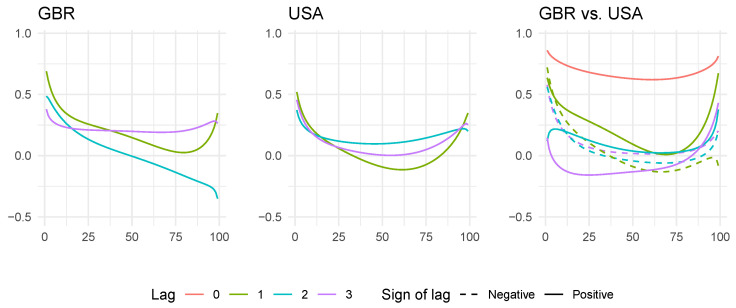
The local auto- and cross-correlations for the UK and the US data, plotted as a function of the percentile of the data along the diagonal $x_{1}=x_{2}$ in the sample space. In the right-hand cross-correlation plot, dashed lines display negative lags, indicating lead–lag relationships from the UK to the US, while positive lags indicate potential lead–lag relationships in the other direction.

**Figure 5 entropy-24-00378-f005:**
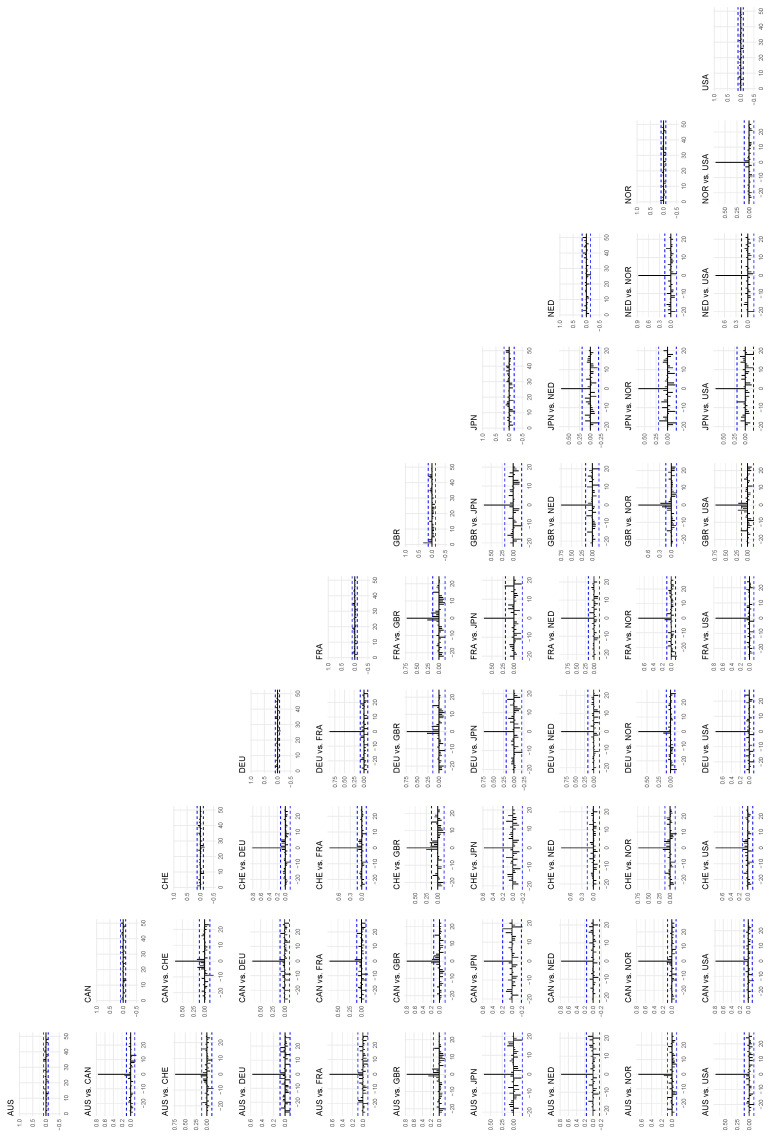
The *global* autocorrelations of the time series in the data set on the diagonal, and the *global* cross-correlations between the time series on the off-diagonal. The *x*-axis indicates lags, and the auto- and cross-correlations are mapped to the *y*-axis.

**Figure 6 entropy-24-00378-f006:**
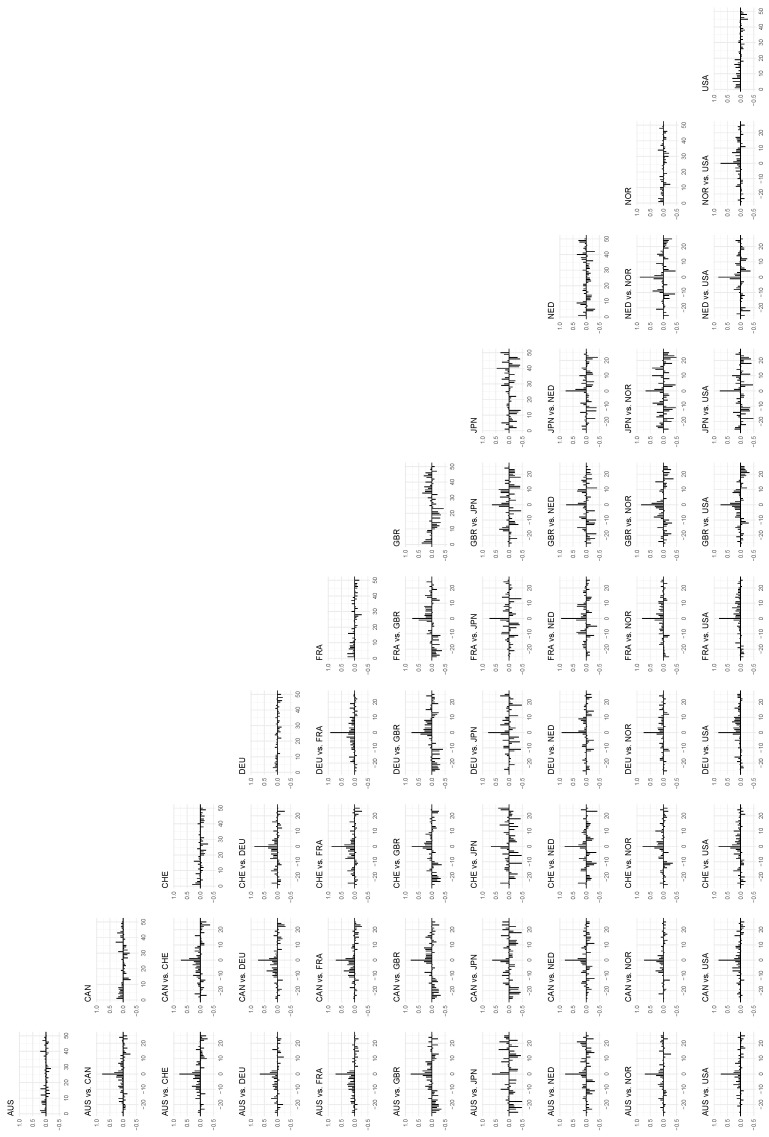
The *local* autocorrelations of the time series in the data set on the diagonal, and the *local* cross-correlations between the time series on the off-diagonal. The *x*-axis indicates lags, and the auto- and cross-correlations are mapped to the *y*-axis. Percentile: 10, bandwidth: 1.

**Figure 7 entropy-24-00378-f007:**
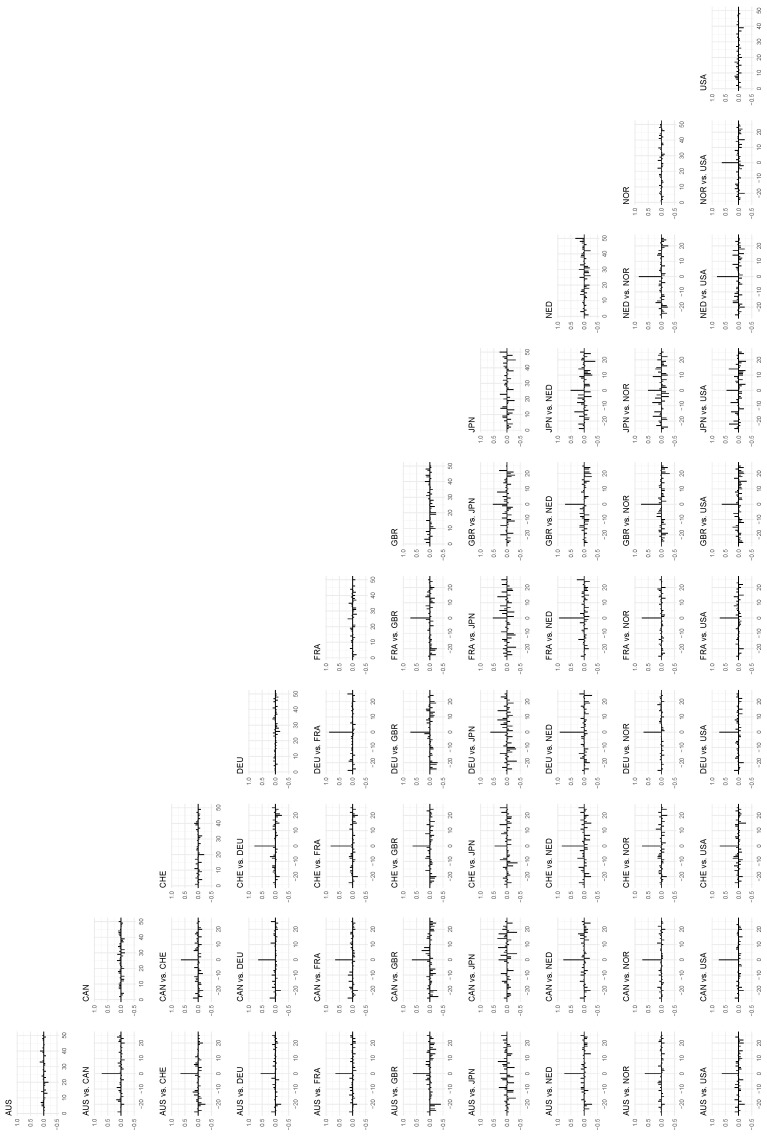
The *local* autocorrelations of the time series in the data set on the diagonal, and the *local* cross-correlations between the time series on the off-diagonal. The *x*-axis indicates lags, and the auto- and cross-correlations are mapped to the *y*-axis. Percentile: 50, bandwidth: 1.

**Figure 8 entropy-24-00378-f008:**
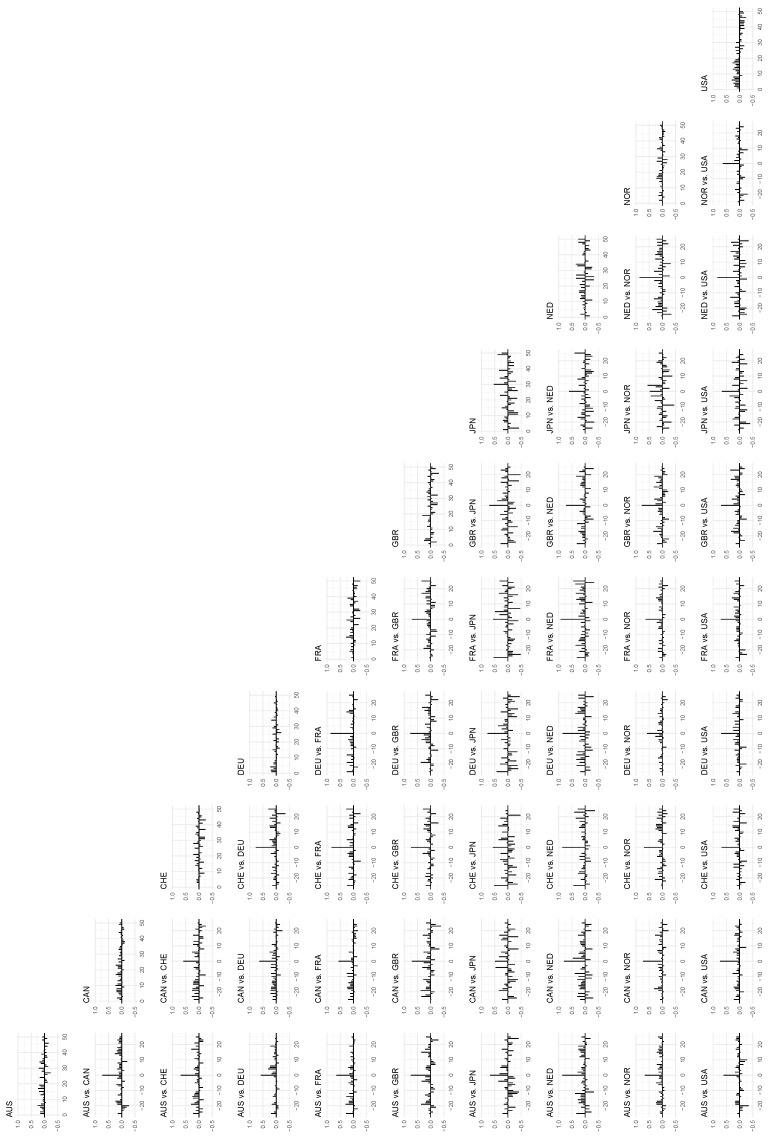
The *local* autocorrelations of the time series in the data set on the diagonal, and the *local* cross-correlations between the time series on the off-diagonal. The *x*-axis indicates lags, and the auto- and cross-correlations are mapped to the *y*-axis. Percentile: 90, bandwidth: 1.

**Figure 9 entropy-24-00378-f009:**
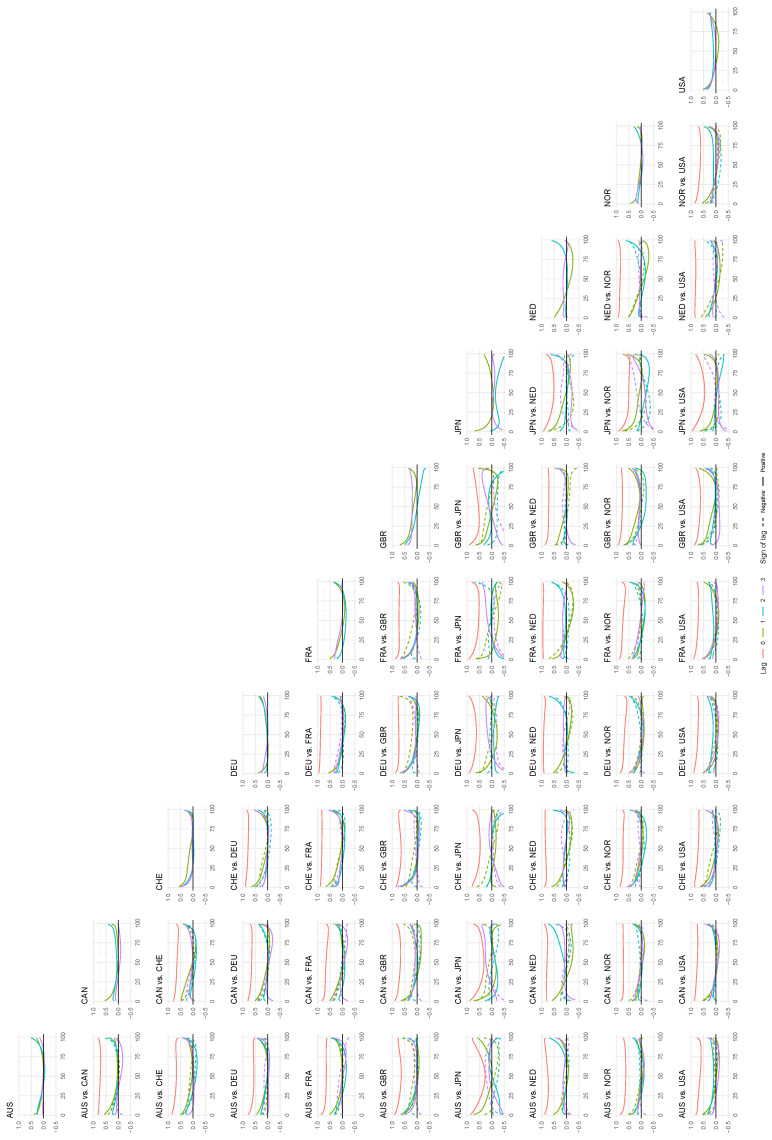
The *local* autocorrelations of the time series in the data set on the diagonal, and the *local* cross-correlations between the time series on the off-diagonal. Plotted as a function of the percentile for the first five lags. Bandwidth: 1.

**Figure 10 entropy-24-00378-f010:**
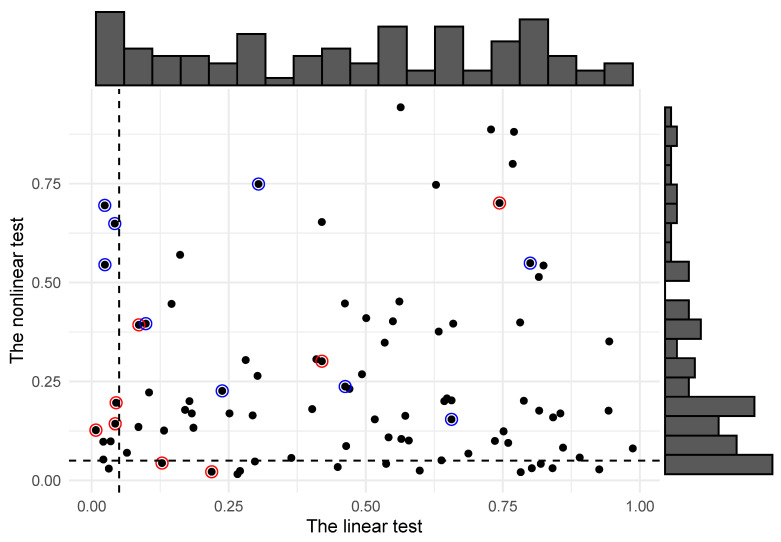
The *p*-values from the tests of Granger noncausality between all pairs of countries in our data set. On the *x*-axis, we have the *p*-values from the linear test, while on the *y*-axis, we have the *p*-values from the nonlinear test based on the local Gaussian correlation. The marginal histograms indicate that the linear test gives *p*-values that are close to uniform. Tests of Granger noncausality from the UK to the other countries in the data set are indicated with a red circle, while the tests of causality from other countries to Norway are indicated with a blue circle.

**Table 1 entropy-24-00378-t001:** Monthly observations on the total market return.

Country	Index	Start Month	# Months
Australia	S&P/ASX 200	1 February 1980	502
Canada	S&P/TSX Composite	1 April 1988	404
France	CAC 40	1 February 1988	406
Germany	DAX 40	1 February 1971	610
Japan	Nikkei 225	1 March 2013	105
Norway	Oslo Stock Exchange, Benchmark Index	1 February 1983	466
Switzerland	SIX Swiss Exchange, Swiss All Share	1 March 1999	273
The Netherlands	Euronext Amsterdam, AEX Index	1 May 2009	151
The United Kingdom	FTSE 100	1 July 2005	197
The United States	S&P 500	1 November 1989	385

**Table 2 entropy-24-00378-t002:** Results for tests of first-order Granger causality, *from* the country in the row *to* the country in the column. The top symbol indicates the result from the linear test, and the bottom symbol indicates the result from the test based on the local Gaussian correlation. Significance codes: - (>0.1) • (0.05) ∗ (0.01) ∗∗ (0.001) ∗∗∗ 0.

	AUS	CAN	CHE	DEU	FRA	GBR	JPN	NED	NOR	USA
Australia (AUS)		∗ •	- -	- -	- -	- ∗	- -	- -	∗ -	- ∗
Canada (CAN)	- ∗		- -	- -	- •	- -	- -	- -	- -	- ∗
Switzerland (CHE)	- •	∗ •		- -	- ∗	- -	- -	- -	- -	∗ •
Germany (DEU)	• •	- -	- -		- -	- ∗	- -	- -	∗ -	- -
France (FRA)	• -	∗ ∗	- -	- -		- -	- -	- -	• -	- -
United Kingdom (GBR)	- ∗	∗∗ -	∗ -	• -	∗ -		- -	- -	∗ -	- ∗
Japan (JPN)	- ∗	- ∗	- -	- -	- -	- -		- •	- -	- -
The Netherlands (NED)	- •	- ∗	- -	- -	- •	- -	- -		- -	- -
Norway (NOR)	- •	- -	- •	- -	- -	- •	- -	- -		- -
United States (USA)	- -	- -	- •	- -	- ∗	- ∗	- -	- -	- -	

## Data Availability

Replication files are available upon request from the corresponding author.

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
