# Peer review of "Local Lead–Lag Relationships and Nonlinear Granger Causality: An Empirical Analysis"

_entropy, 2022, doi:10.3390/e24030378_

Round 1
Reviewer 1 Report
This paper proposes a nonlinear Granger causality test for analyzing the local dynamic relationships in in time series as applied to monthly financial indices across several countries. The approach is an interesting one and is very useful for thinking about how dynamics my change across the distribution of the time series. I just have several suggestions that might help convey the usefulness and value of the approach.
- The figures are generally quite difficult to read given the that they generally extend across arbitrarily large ranges. For example, Figure 1 probably shouldn't extend below -0.1 or above 0.5. Similarly with Figure 2 which could be even narrow range. Narrowing the ranges on the Figures where possible might highlight the dynamics much better.
- I think it is probably worth discuss and emphasizing how to interpret the implications of Granger causality (or the lack thereof) when only looking at one aspect of the distribution rather than the entire. I think it is important to be very clear on how to think about this approach.
- The model considered in section 3 has specific stationarity conditions that are assumed to be satisfied. At the same time, Figure 3 displays fairly persistent correlations in the tails of the distributions. Can we be certain that the data satisfies the stationarity conditions? Both globally and locally? What are the implications for this analysis if these stationarity assumptions are not satisfied locally?
Author Response
We thank the reviewer for their careful reading of the manuscript. We respond to the comments in a point-by-point fashion below.
"This paper proposes a nonlinear Granger causality test for analyzing the local dynamic relationships in in time series as applied to monthly financial indices across several countries. The approach is an interesting one and is very useful for thinking about how dynamics my change across the distribution of the time series. I just have several suggestions that might help convey the usefulness and value of the approach."
Our response: Thank you. Please see our response to your comments below.
"The figures are generally quite difficult to read given the that they generally extend across arbitrarily large ranges. For example, Figure 1 probably shouldn't extend below -0.1 or above 0.5. Similarly with Figure 2 which could be even narrow range. Narrowing the ranges on the Figures where possible might highlight the dynamics much better."
Our response: We agree that the scales of Figures 1 and 2 are not ideal for reading the finer details of the autocorrelation functions. Nevertheless, these plots should be read together since the scale on the y-axis is comparable across the figures. We have therefore chosen to leave the Figures as they are. The main message in Figure 1, for example, is that 5 of 6 plots show no significant autocorrelation, while the final plot does.
"I think it is probably worth discuss and emphasizing how to interpret the implications of Granger causality (or the lack thereof) when only looking at one aspect of the distribution rather than the entire. I think it is important to be very clear on how to think about this approach."
Our response: We have clarified this aspect in the discussion following equation (4), and we have added reference to this discussion at the end of the penultimate paragraph on page 3.
"The model considered in section 3 has specific stationarity conditions that are assumed to be satisfied. At the same time, Figure 3 displays fairly persistent correlations in the tails of the distributions. Can we be certain that the data satisfies the stationarity conditions? Both globally and locally? What are the implications for this analysis if these stationarity assumptions are not satisfied locally?"
Our response: It is a standard assumption that stock return series are stationary, even though they are known to possess long memory properties. It is an important assumption, and we have added it explicitly to the text at the beginning of Section 4.
Reviewer 2 Report
The authors present an analysis of monthly time series of financial data of stock market indices from various countries, to illustrate the applicability and effectiveness of the method of local Gaussian approach in detecting causal links between the series of the dataset. The distribution of the data is approximated by Gaussians, that are derived from fits in particular "areas" of the variable values, e.g. one close to the left tail of the distribution, another around the median and another close to the right tail, etc, which permit a more local modelling of the underlying statistical process and allow for focused analyses of the interplay between the variables. The analysis supports the claim that the method is more sensitive than simple cross-correlation analysis and can thus detect causal relations that would otherwise remain hidden.
The method is very interesting, as it allows the use of the full arsenal of the multivariate Gaussian formalism even in cases where the data follow a different distribution and the Granger causality tests can be applied to detect non-linear relations between variables, something which is useful far beyond the scope of financial data. The main issues is that the work relies heavily on past works by the same authors and thus, some points are not as clearly defined as I, and I believe many readers, would want them to be. With some additional text and slightly improved figures, it would make a fine addition to the Entropy journal.
Some comments/questions:
Section 1.
A. The choice of monthly data is somewhat perplexing in the context of this work. Especially since the local fits are defined in terms of percentiles, I would expect that the results would be even more pronounced in the case of weekly or daily data. Is there any other reason for the choice of this particular time cadence? Are daily data not readily available? Is the process too time consuming for longer time series?
Section 2.1
B. I would like some additional discussion on the method, focusing on how good is this approximation for the case of non-gaussian distributions, as this is one of the most critical aspects of this work. If the global distribution is e.g. skewed, is this type of analysis still valid? What are its limitations? Would you use a smaller 'b' value for more complex distributions? How do you ensure that the approximation holds?
C. What is the kernel K in equation (1)? Is it a simple step function of width 'b'? I don't thing it is defined in the text.
D. Is the 'b' defined in the space of the original variables X, or their transformations Z?
Section 2.2
E. I believe some more text explaining the definition of the test would be useful. The null hypothesis is defined upon a linear relationship between the variables (and their time lags) according to Granger's formalism. How is the test statistic defined as a "distributional nonlinear test". I know the authors point to Otneim and Tjøstheim (2021), but since their previous works were mostly published on Economincs-related journals I do not think that many of Entropy's readers will be familiar with the concepts outlined here. Please expand this point a little more.
Section 3
F. Would this example work if the ARCH model used a different distribution for the epsilon_t, rather than the Gaussian? This goes back to question B. If you could run the simulation with a different underlying distribution and show the applicability of the method in this case, I think you would have a much stronger argument in support of your work.
G. Please add labels on the x and y axes of the plots, here and in the next sections.
H. Why are the confidence intervals only shown on the global plots?
I. Have you tried examining these relations using the framework of Mutual Information and/or Conditional Mutual Information measures? Granger causality can also be defined using these measures, which are by their very nature non-linear.
Author Response
"The authors present an analysis of monthly time series of financial data of stock market indices from various countries, to illustrate the applicability and effectiveness of the method of local Gaussian approach in detecting causal links between the series of the dataset. The distribution of the data is approximated by Gaussians, that are derived from fits in particular "areas" of the variable values, e.g. one close to the left tail of the distribution, another around the median and another close to the right tail, etc, which permit a more local modelling of the underlying statistical process and allow for focused analyses of the interplay between the variables. The analysis supports the claim that the method is more sensitive than simple cross-correlation analysis and can thus detect causal relations that would otherwise remain hidden.
The method is very interesting, as it allows the use of the full arsenal of the multivariate Gaussian formalism even in cases where the data follow a different distribution and the Granger causality tests can be applied to detect non-linear relations between variables, something which is useful far beyond the scope of financial data. The main issues is that the work relies heavily on past works by the same authors and thus, some points are not as clearly defined as I, and I believe many readers, would want them to be. With some additional text and slightly improved figures, it would make a fine addition to the Entropy journal."
Our response: The reviewer understands the idea behind this work precisely. We would like to thank the reviewer for his/her careful comments, which we address in a point-by-point fashion below.
"Some comments/questions:
Section 1.
A. The choice of monthly data is somewhat perplexing in the context of this work. Especially since the local fits are defined in terms of percentiles, I would expect that the results would be even more pronounced in the case of weekly or daily data. Is there any other reason for the choice of this particular time cadence? Are daily data not readily available? Is the process too time consuming for longer time series? "
Our response: The reviewer is entirely correct. The results are much more clear when using daily data. There are mainly two reasons for this choice: Our main reference, Rapach, Strauss, and Zhou (2013), use monthly data in their analysis of Granger causality. Furthermore, daily returns are not directly comparable across different time zones due to different closing times in different countries. This problem warrants detailed adjustments to the data, which is outside the scope of our paper. We have, however, added a comment about this choice in the first paragraph of Section 4.
"Section 2.1
B. I would like some additional discussion on the method, focusing on how good is this approximation for the case of non-gaussian distributions, as this is one of the most critical aspects of this work. If the global distribution is e.g. skewed, is this type of analysis still valid? What are its limitations? Would you use a smaller 'b' value for more complex distributions? How do you ensure that the approximation holds?"
Our response: We have added a paragraph at the top of page 3 addressing this issue.
"C. What is the kernel K in equation (1)? Is it a simple step function of width 'b'? I don't thing it is defined in the text."
Our response: Thank you for noticing. We use the standard normal density as kernel function and have added a comment just after eq. (1) about this.
"D. Is the 'b' defined in the space of the original variables X, or their transformations Z?"
Our response: The bandwidth is determined on the z-scale. We have added a comment about this on page 3.
"Section 2.2
E. I believe some more text explaining the definition of the test would be useful. The null hypothesis is defined upon a linear relationship between the variables (and their time lags) according to Granger's formalism. How is the test statistic defined as a "distributional nonlinear test". I know the authors point to Otneim and Tjøstheim (2021), but since their previous works were mostly published on Economincs-related journals I do not think that many of Entropy's readers will be familiar with the concepts outlined here. Please expand this point a little more."
Our response: Thank you for pointing this out. We have added some paragraphs before equation (4), where we explain the concept of this test in some more detail. See also our response to point I.
"Section 3
F. Would this example work if the ARCH model used a different distribution for the epsilon_t, rather than the Gaussian? This goes back to question B. If you could run the simulation with a different underlying distribution and show the applicability of the method in this case, I think you would have a much stronger argument in support of your work."
Our response: Yes, this example works for a wide variety of data generating processes. We have repeated the experiment with epsilon_t drawn from the heavy-tailed t(5)-distribution. The results are practically identical, so we have chosen not to show the plots in the paper. We have, however, included a comment referring to these results at the end of Section 3 and included two new references Otneim and Tjøstheim (2017,2018), where there is a wider class of distributions, including a skewed t-distribution.
"G. Please add labels on the x and y axes of the plots, here and in the next sections."
Our response: We have experimented with various solutions to this. Adding labels to all the plots makes the figures too cluttered, in our opinion. So instead, we have chosen to add information about the labels in the figure captions.
"H. Why are the confidence intervals only shown on the global plots?"
Our response: The confidence intervals for local plots can be obtained using the block bootstrap. That is, however, quite time-consuming, and since we do not use them to make inferences, we have chosen to omit them in the plots. We refer instead to the formal testing in Section 4, where we obtain p-values directly for the conditional independence tests, also using a bootstrap procedure. We have added a comment about this in Section 3.
"I. Have you tried examining these relations using the framework of Mutual Information and/or Conditional Mutual Information measures? Granger causality can also be defined using these measures, which are by their very nature non-linear."
Our response: We have not used other non-parametric conditional independence tests than the one based on the local Gaussian correlation. We have, however, included a remark just before equation (4) where we point out that there certainly are other ways to do this, including the concept of mutual information.